# Upscaling the Surveillance of Tick-Borne Pathogens in the French Caribbean Islands

**DOI:** 10.3390/pathogens9030176

**Published:** 2020-03-01

**Authors:** Mathilde Gondard, Sabine Delannoy, Valérie Pinarello, Rosalie Aprelon, Elodie Devillers, Clémence Galon, Jennifer Pradel, Muriel Vayssier-Taussat, Emmanuel Albina, Sara Moutailler

**Affiliations:** 1UMR BIPAR, Animal Health Laboratory, ANSES, INRAE, National Veterinary School of Alfort, Paris-Est University, Maisons-Alfort, 94700 Paris, France; mathilde.gondard@gmail.com (M.G.); elodie.devillers@anses.fr (E.D.); clemence.galon@anses.fr (C.G.); Muriel.taussat@inrae.fr (M.V.-T.); 2CIRAD, UMR ASTRE, F-97170 Petit-Bourg, 97170 Guadeloupe, France; valerie.pinarello@cirad.fr (V.P.); rosalie.aprelon@cirad.fr (R.A.); jennifer.pradel@cirad.fr (J.P.); emmanuel.albina@cirad.fr (E.A.); 3IdentyPath Platform, Laboratory for Food Safety, ANSES, Maisons-Alfort, 94700 Paris, France; sabine.delannoy@anses.fr; 4ASTRE, Univ Montpellier, CIRAD, INRAE, 34000 Montpellier, France

**Keywords:** tick, bacteria, parasites, Caribbean, microfluidic real-time PCR

## Abstract

Despite the high burden of vector-borne disease in (sub)tropical areas, few information are available regarding the diversity of tick and tick-borne pathogens circulating in the Caribbean. Management and control of vector-borne disease require actual epidemiological data to better assess and anticipate the risk of (re)emergence of tick-borne diseases in the region. To simplify and reduce the costs of such large-scale surveys, we implemented a high-throughput microfluidic real-time PCR system suitable for the screening of the main bacterial and parasitic genera involved in tick-borne disease and potentially circulating in the area. We used the new screening tool to perform an exploratory epidemiological study on 132 adult specimens of *Amblyomma variegatum* and 446 of *Rhipicephalus microplus* collected in Guadeloupe and Martinique. Not only the system was able to detect the main pathogens of the area—*Ehrlichia ruminantium*, *Rickettsia africae*, *Anaplasma marginale*, *Babesia bigemina* and *Babesia bovis*—but the system also provided evidence of unsuspected microorganisms in Caribbean ticks, belonging to the *Anaplasma*, *Ehrlichia*, *Borrelia* and *Leishmania* genera. Our study demonstrated how high-throughput microfluidic real-time PCR technology can assist large-scale epidemiological studies, providing a rapid overview of tick-borne pathogen and microorganism diversity, and opening up new research perspectives for the epidemiology of tick-borne pathogens.

## 1. Introduction

Among hematophagous arthropods, ticks transmit the greatest variety of pathogens of public health and veterinary importance whose incidence is growing worldwide [1]. The French West Indies, including the islands of Guadeloupe and Martinique, are located in the heart of the Caribbean Neotropical zone, a cosmopolitan area characterized by a tropical climate, intercontinental trade and animal movements (legal and illegal trade as well as bird migration) that are favorable for the introduction and spread of ticks and tick-borne pathogens (TBPs) [2]. Yet, the epidemiological situation of the Caribbean area with regard to the diversity of tick species and tick-borne diseases (TBDs) is poorly documented [3].

*Amblyomma variegatum, Rhipicephalus microplus* and *Rhipicephalis sanguineus* sensu lato are the main tick species found in the French Antilles that are involved in the transmission of TBPs of medical and veterinary importance [3]. While *Rhipicephalis sanguineus* sensu lato are mainly found infesting dogs, *Amblyomma variegatum*, also known as the tropical bont tick (TBT) in the Caribbean, and *Rhipicephalus microplus* (the “cattle tick”) have been the two main tropical livestock pests since their introduction in the Caribbean through imports of infested animals from Africa and Asia in the 18th–19th centuries [3,4,5,6,7,8,9].

*R. microplus*, a one-host tick highly specific to cattle, is mainly involved in the transmission of *Anaplasma marginale*, *Babesia bovis* and *Babesia bigemina*, causing bovine anaplasmosis and babesiosis, respectively. These endemic pathogens are responsible for important economic loss to farming industries in the Caribbean and are still a sanitary threat [7,10].

*A. variegatum* is a three-host tick species, with immature stages that can parasitize a wide range of hosts, including rodents, mongooses and birds, as well as an adult stage that is more specific to cattle [11]. This tick species is mainly involved in *Ehrlichia ruminantium* transmission, the causative agent of heartwater, a fatal ruminant ehrlichiosis. Although *A. variegatum* is present in both Martinique (mainly in the south) and Guadeloupe (widespread), *E. ruminantium* has only been reported in Guadeloupe [12]. In addition, *A. variegatum* ticks are also a vector of *Rickettsia africae*, which is common in the Caribbean and can induce human rickettsiosis, called African tick-bite fever [9,13,14]. African tick-bite fever remains a concern mainly for travelers. Indeed, despite high levels of tick infection and seroprevalence in human and cattle sera, only two human cases of African tick-bite fever have been reported to date, only in travelers returning from Guadeloupe [9,15]. Lastly, *A. variegatum* is also involved in the epidemiology of *Theileria mutans* and *Theileria velifera*, two cattle parasites with low and no virulence, respectively [6,8]. However, very few information is available on the distribution and prevalence of these two Apicomplexa in the Caribbean.

Most of the epidemiological data available did not survey or determine the diversity of TBPs circulating in the Caribbean, since they were often limited to the detection of some well-known pathogens, via serological studies in animals or humans, or on molecular biology testing (PCR, nested PCR) [16,17]. Thus, regarding the lack of recent information and the limited extent of the epidemiological data available, new insight into the epidemiology of ticks and TBPs was needed to better address the prevalence and (re)emergence of TBDs in the Caribbean.

In order to improve the surveillance ability of tick-borne pathogens in the Neotropical area, we implemented a new large-scale screening tool based on a microfluidic real-time PCR approach. Microfluidic real-time PCR is based on the use of microfluidic chips allowing the performance of up to 9216 individual PCR reactions per run, and thus the simultaneous detection of up to 96 targets in up to 96 samples. The recent development and use of a microfluidic real-time PCR for the rapid and concomitant detection of a large panel of TBPs in European ticks has paved the way for promising and broader surveillance capacities [18,19,20,21,22]. Here, we adapted and designed a new microfluidic real-time PCR system suited to the simultaneous screening of the main bacteria and protozoans potentially transmitted by ticks in the Caribbean. Not only did the system enable the direct detection of 49 bacterial and parasitic species, but it also enabled, within a single experiment, broader capacities for the surveillance of potentially pathogenic microorganisms by targeting the main bacterial and protozoan genera involved in human and animal vector-borne diseases (one protozoan phylum and eight bacterial and protozoan genera). In addition, the system enabled the molecular identification of the three well-known tick species involved in TBDs in the Caribbean in order to confirm the morphological tick species identification determined on the field. Finally, we used the new high-throughput detection tool to conduct large-scale screening of TBPs in 132 *A. variegatum* and 446 *R. microplus* adult specimens collected in Guadeloupe and Martinique. We demonstrated the system’s ability to detect well-known TBPs occurring in the French West Indies, as well as unsuspected TBPs and potential new microorganisms. This new method can considerably improve the ability to monitor emerging and non-emerging TBPs through large-scale surveys in the Caribbean area.

## 2. Results

### 2.1. Implementation of the High-Throughput Microfluidic Real-Time PCR System for Tick-Borne Pathogen Screening

The high-throughput microfluidic real-time PCR system developed for the screening of known and potential TBPs in Caribbean ticks included 61 sets of primers and probes. Among them, 49 designs were developed for the detection of bacterial (n = 32) and protozoan (n = 17) species and bacterial (n = 5) and protozoan (n = 3) genera/phyla (Table 1). Three sets of primers and probes were developed for the molecular identification of the three tick species found in the Caribbean: *A. variegatum*, *R. microplus* and *R. sanguineus* sensu lato (Table 1). Lastly, we developed a design targeting a conserved region of the 16S rRNA genes in ticks, called “Tick spp.”, used as a control for DNA/RNA extraction (Table 1).

The detection ability of each design and the effect of pre-amplification on detection signals were first checked by TaqMan real-time PCR on a LightCycler 480 apparatus using a range of dilutions of positive controls (Table 1, Appendix A). Three kinds of positive controls were used, including bacterial or protozoan cultures when available, DNA from infected ticks or blood samples, and plasmidic constructions as a last resort (Table 1). Except for the design targeting Borrelia burgdorferi sensu stricto, which never succeeded in detecting the positive controls even after a pre-amplification step, the remaining 60 designs targeting TBPs and tick species were able to detect their target with Ct values between 6 and 38 (data not shown). Pre-amplification improved the quality of detection and was, therefore, validated as part of the screening protocol (see Appendix A). The relative specificity of the 61 designs was then evaluated using the BioMark system and a total of 62 positive controls (Figure 1, Appendix A).

Forty-three primer/probe sets were able to specifically detect and amplify their target using a Ct cut-off value of 30; they were then directly validated (Figure 1). The remaining designs were able to detect and amplify their target, but they also gave positive results in outgroup controls. Interestingly, two kinds of unsuspected signals were observed: some were related to cross-reactions with closely related species and some to potential co-infections in controls corresponding to field samples (Figure 1). Thus, eight designs—*Rickettsia massiliae*, *Rickettsia conorii*, *Bartonella henselae*, *Bartonella bacilliformis*, *Babesia canis vogeli*, *Babesia microti*, *Theileria parva* and *Hepatozoon americanum*—gave positive results in outgroup controls, revealing cross-reactions with one to two closely related species (Figure 1). Caution will be required when interpreting results obtained with these designs. Seven designs—*Rickettsia* spp., *Rickettsia felis*, *Rickettsia africae*, *Apicomplexa*, *Babesia bigemina*, *Hepatozoon* spp. and *Hepatozoon canis*—gave positive results in outgroup controls linked to potential co-infection in controls corresponding to DNA from infected ticks or blood samples (Figure 1). As co-infections may occur in natural tick or blood samples, these unexpected detections in biological samples were likely due to the natural (co)occurrence of microorganisms rather than to cross-reactions. Finally, the *Babesia ovis* and *Rickettsia rickettsii* designs gave multiple cross-reactions with closely related species or distant outgroups and, thus, were considered as non-specific and removed from the rest of the study (Figure 1). More details on the relative specificity analysis of the designs are available in Appendix B.

To conclude, with the exception of the sets of primers and probes targeting *Borrelia burgdorferi* sensu stricto, *Babesia ovis* and *Rickettsia rickettsii* that were ultimately removed from the study, the 58 remaining designs were validated for the high-throughput screening of pathogens in Caribbean ticks, taking into account the notified cross-reactions.

### 2.2. Large-Scale TBP Detection Survey in Ticks from Guadeloupe and Martinique

A total of 578 adult ticks were collected from cattle in Guadeloupe and Martinique. In total, 523 samples were tested using the BioMarkTM system developed in this study. The molecular identification of *Amblyomma variegatum* and *Rhipicephalus microplus* using the corresponding specific designs were consistent with the morphological identification made after tick collection. The number of positive ticks and the corresponding infection rates for each detected pathogen were calculated for 132 *A. variegatum* as well as 165 and 281 *R. microplus* specimens from Guadeloupe and Martinique, respectively (Figure 2). As some of the *R. microplus* samples corresponded to pools of two to four adult specimens, we reported the minimum and maximum infection rates (see Materials and Methods).

Conventional PCRs/nested PCRs followed by amplicon sequencing were performed on several tick samples presenting low Ct values to confirm the results of the newly designed BioMark^TM^ system (see Materials and Methods section). Identity percentages of the sequences obtained with reference sequences available in GenBank (NCBI) are presented in Table 2.

#### 2.2.1. Detection of Known TBPs in Caribbean Ticks 

Seven TBPs known to circulate in the Caribbean were detected in ticks from Guadeloupe and Martinique: *R. africae*, *E. ruminantium*, *A. marginale*, *B. bigemina*, *B. bovis*, *T. mutans* and *T. velifera* (Figure 2).

*Rickettsia* spp. were only detected in ticks collected in Guadeloupe (Figure 2). *R. africae* was identified in 95.5% of the *A. variegatum* samples (Figure 2). In contrast, *Rickettsia* spp. detected in 15.2%–23% of the *R. microplus* samples from Guadeloupe were not directly identified as *R. africae* with the BioMark^TM^ system (Figure 2). Thus, 14 *A. variegatum* (6/14) and *R. microplus* (8/14) samples positive for *Rickettsia* spp. were tested by nested PCR with primers targeting the ompB gene; this was followed by sequencing. All the sequences recovered were identical and displayed 100% identity with *R. africae*, confirming that the *Rickettsia* spp. detected in *R. microplus* from Guadeloupe corresponded also to *R. africae*. (Table 2). The consensus sequence was deposited under the name *Rickettsia africae* Tick208 (accession number MK049851).

*E. ruminantium* was identified in 5.3% of the *A. variegatum* ticks from Guadeloupe (Figure 2). We confirmed the presence of *E. ruminantium* nucleic acids by testing one sample of *A. variegatum* by conventional PCR targeting the 16S rRNA genes; this was followed by amplicon sequencing. The sequence obtained displayed 100% sequence identity with *E. ruminantium* and was deposited under the name *Ehrlichia ruminantium* Tick116 (accession number MK049848) (Table 2).

*A. marginale* was identified in *R. microplus* ticks from both islands, with infection rates reaching 3%–4.2% and 39.9%–41.3% of specimens from Guadeloupe and Martinique, respectively (Figure 2). We confirmed the detection of *A. marginale* by testing two samples of *R. microplus* by conventional PCR targeting the 16S rRNA genes; this was followed by amplicon sequencing. We obtained two identical sequences, deposited under the name *Anaplasma* sp. Tick283 (accession number MK049844), which displayed 100% sequence identity with *Anaplasma* spp. including *A. marginale* (Table 2).

*B. bigemina* was detected in 0.6%–1.2% and 12.5%–12.8% of the *R. microplus* ticks from Guadeloupe and Martinique, respectively (Figure 2). *B. bovis* was only detected in ticks from Martinique, with an infection rate of 0.7% in *R. microplus* samples (Figure 2). As conventional and nested PCR did not succeed in detecting these parasites, we directly sequenced amplicons obtained with the *B. bigemina* and *B. bovis* designs developed here, and corresponding sequences were identified (accession numbers MK071738 and MK071739 respectively) (Table 2).

*T. velifera* and *T. mutans* were detected in both tick species and on both islands. *T. velifera* was identified in 42.3% of the *A. variegatum* samples and in 23.6%–31.5% and 25.6%–26% of the *R. microplus* samples from Guadeloupe and Martinique, respectively (Figure 2). Moreover, *T. mutans* was detected in 1.5% of the *A. variegatum* samples and in 1.8%–2.4% and 1.4% of the *R. microplus* samples from Guadeloupe and Martinique, respectively (Figure 2). Unfortunately, neither conventional PCR nor BioMark amplicon sequencing succeeded in confirming the BioMark results.

#### 2.2.2. Detection of Unexpected Microorganisms in Caribbean Ticks 

Unexpected signals were obtained during the screening of microorganisms in ticks from Guadeloupe and Martinique, including the first detection of untargeted species belonging to the genera *Anaplasma*, *Ehrlichia*, *Borrelia* and *Leishmania* (Figure 2).

*Ehrlichia* spp. were detected in *R. microplus* ticks from both islands, with infection rates reaching 4.2%–6.6% and 47.7%–49.1% in Guadeloupe and Martinique, respectively (Figure 2). We tested two of the *Ehrlichia* spp.-positive *R. microplus* samples by conventional PCR targeting the 16S rRNA genes in order to identify the *Ehrlichia* spp. present in the Caribbean sample. We obtained two identical sequences, deposited under the name *Ehrlichia* sp. Tick428 (accession number MK049849) (Table 2). Phylogenetic and genetic distance analyses were performed using a portion of the 16S rRNA genes of several *Ehrlichia* species (Figure 3). The *Ehrlichia* sp. Tick428 sequence was found within a cluster including various uncharacterized *Ehrlichia* species detected in ticks from Asia and Africa (Figure 3).

In addition, in around 50% (at least 4/8 ticks) and 18% (at least 22/114 ticks) of the *R. microplus* specimens positive for *Anaplasma* spp., none of the *Anaplasma* species targeted by the BioMark^TM^ system gave signals, suggesting the presence of an unexpected or new *Anaplasma* spp. (Figure 2). We tested two of the *Anaplasma* spp.-positive *R. microplus* samples by conventional PCR targeting the 16S rRNA genes. We obtained two identical sequences, deposited under the name *Anaplasma* sp. Tick314 (accession number MK049845) (Table 2). This sequence displayed 100% sequence identity with *Candidatus* Anaplasma boleense. Phylogenetic and genetic distance analyses were performed using a portion of the 16S rRNA genes of several *Anaplasma* species (Figure 4). The *Anaplasma* sp. Tick314 sequence was found in a cluster including *Candidatus* Anaplasma boleense, *Anaplasma platys* and *Anaplasma phagocytophilum*.

*Borrelia* spp. were detected in both tick species from both islands (Figure 2). Infection rates reached 5.3% in *A. variegatum* and 0.6% and 4.3% in *R. microplus* from Guadeloupe and Martinique, respectively (Figure 2). None of the specific targeted *Borrelia* species causing Lyme disease (*Borrelia burgdorferi* sensu lato), or the *Borrelia* relapsing fever group, gave any positive results, suggesting the occurrence of a new or unexpected *Borrelia* spp. in our samples (Figure 2). We tested 30 of the *Borrelia* spp.-positive ticks by nested PCR targeting the flaB genes. Interestingly, we obtained two sequences according to the tick species analyzed. The *Borrelia* sp. Tick7 (accession number MK049846) sequence was recovered from one *A. variegatum* sample from Guadeloupe, and the *Borrelia* sp. Tick457 sequence (accession number MK049847) was recovered from four *R. microplus* samples from Martinique (Table 2). Phylogenetic and genetic distance analyses were performed using a portion of the flaB gene of several *Borrelia* species (Figure 5). Surprisingly, the *Borrelia* sp. Tick7 sequence recovered from the *A. variegatum* sample, and found to be closely related to *B. anserina*, displayed an intermediate position, sharing homology with both the relapsing fever and Lyme disease groups (Figure 5). Lastly, the *Borrelia* sp. Tick457 sequence recovered from the *R. microplus* samples confirmed the previous observations, forming a cluster with various relapsing fever *Borrelia* species encountered in hard ticks, including *B. lonestari* and *B. theileri* (Figure 5).

Lastly, 0.7% of the *R. microplus* ticks from Martinique were positive for *Leishmania* spp. (Figure 2). We tested two of the *Leishmania* spp.-positive ticks by nested PCR targeting the small subunit rRNA gene. We obtained one sequence from one sample, deposited under the name *Leishmania martiniquensis* Tick389 (accession number MK049850) (Table 2). This sequence displayed 100% identity with both the *Leishmania martiniquensis* and *Leishmania siamensis* sequences (Table 2).

#### 2.2.3. Co-Infections in Ticks in Guadeloupe and Martinique

We analyzed the co-infections observed in *Amblyomma variegatum* (n = 132 samples), *Rhipicephalus microplus* collected in Guadeloupe (n = 116 samples, including individual and pooled specimens) and Martinique (n = 275 samples, including individual and pooled specimens). In Guadeloupe, almost all of the *A. variegatum* samples (99.2%) were infected with at least one pathogen, whereas only 56% of the *R. microplus* samples were infected (Figure 6, Table A3). In contrast, 81% of the *R. microplus* from Martinique were infected with at least one pathogen (Figure 6, Table A3). High and similar percentages of the two tick species were infected with either one or two pathogens. The percentages drastically dropped for co-infection with three pathogens, with less than 10% of the ticks infected. Respectively one and nine *A. variegatum* and *R. microplus*, from Guadeloupe and Martinique, were co-infected with four pathogens, and one *R. microplus* from Martinique was found infected with five pathogens (Figure 6, Table A3).

*A. variegatum* from Guadeloupe were find heavily infected by *R. africae,* yet it did not seem to affect the presence of other pathogen/microorganisms that were all found in co-infection with the bacteria (Table A4). Interestingly, in *R. microplus* from Guadeloupe, most of the single-infection reported corresponded to *R. africae* (12.9%) or *T. velifera* (21.6%) (Table A5). Positive association have been identified between *T. velifera* and *T. mutans*, and *Anaplasma* spp./*Borrelia* spp. (Table A5). Finally, in *R. microplus* from Martinique, five positive associations have been detected, including *T. mutans*/*T. velifera*, *T. mutans/Leishmania* spp., *T. mutans/Borrelia* spp., *T. velifera*/*B. bigemina* and *A. marginale*/*Ehrlichia* spp. (Table A6). The result of the co-occurrence test should be taken with caution and deserves further investigation regarding the few number of positive samples (Table A5, A6). Nevertheless, no exclusion seemed to occur between the pathogens/microorganisms detected in the two tick species from Guadeloupe and Martinique. More details on co-infections in ticks from Guadeloupe and Martinique are available in Appendix C.

## 3. Discussion

In this study, a high-throughput microfluidic real-time PCR system based on the use of multiple primers/probes was developed for large-scale surveys of bacteria and protozoans potentially transmitted by ticks from the Caribbean area. The association of genus and species primer/probe designs targeting TBPs improved the technology’s screening capacity, enabling not only the identification of infectious agents known to circulate in the studied area, but also the detection of unsuspected TBPs and new microorganisms belonging to the main bacterial and protozoan genera/phyla involved in TBDs worldwide. Nevertheless, as some endosymbiotic microorganisms may belong to known TBP genera, such as *Rickettsia* and *Coxiella*, confirmatory tests are required before suggesting the presence of a pathogenic microorganism [23,24,25]. When analyzing the specificity of the microfluidic real-time PCR system, cross-reactions were observed for some designs targeting closely related species; these must be taken into account when interpreting the results. Due to high design constraints and a lack of available sequences in public databases, the improvement of such cross-reacting oligonucleotides remains challenging. Here, the concomitant use of bacterial and protozoan genera can assist in identifying non-specific signals. In addition to detecting microorganisms, we developed sets of primers and probes enabling the molecular identification of the three main tick species involved in TBDs in the Caribbean: *A. variegatum*, *R. microplus* and *R. sanguineus* s.l. As the morphological identification of ticks collected in the field remains challenging, molecular identification can be used to confirm the identification of the tick species analyzed [16,26,27].

We used the newly developed high-throughput microfluidic real-time PCR system to perform an exploratory epidemiological study on TBPs and microorganisms potentially circulating in *A. variegatum* and *R. microplus* ticks collected on cattle in Guadeloupe and Martinique. The analysis provided an overview of the diversity of microorganisms belonging to the main bacterial and protozoan genera potentially transmitted by ticks. It enabled the detection both of known TBPs of public and animal health importance in the area that require surveillance and of unexpected microorganisms occurring in Caribbean ticks.

The four main pathogens responsible for ruminant diseases in the Caribbean, currently classified as notifiable diseases by the World Organization for Animal Health (OIE), have been detected by the microfluidic real-time PCR system. These are *E. ruminantium* in *A. variegatum* specimens and *A. marginale*, *B. bigemina* and *B. bovis* in *R. microplus*.

Interestingly, the *E. ruminantium* infection rate in *A. variegatum* reported in our study was much lower compared to in previous studies conducted between 2003 and 2005 in Guadeloupe (5.1% versus 36.7%) [12]. Although different study designs were used (different sampling strategies, study periods, detection methods, etc.), which may explain this difference, it would be worth further investigating whether the tick infection rate for *E. ruminantium* has decreased in Guadeloupe and possibly assessing the epidemiological impact in terms of the incidence and prevalence of heartwater in the ruminant population. These results are all the more surprising since systematic TBT surveillance and control programs have been discontinued in the French Antilles following the end of the POSEIDOM (Specific Options Program due to the remoteness and insularity of the overseas departments) eradication programs in 2006.

In this study, we have documented infection rates for *B. bigemina*, *B. bovis* and *A. marginale* in the *R. microplus* vector tick in the French West Indies for the first time. Indeed, records of such pathogens are mostly based on seroprevalence studies in cattle [7,8,10].

*R. microplus* ticks are both vectors and reservoirs of *B. bigemina* and *B. bovis*, transmitting the parasites transovarially and trans-stadially [28,29]. As *R. microplus* ticks and cattle are both reservoirs of infection, the infection rates reported here seemed quite low. The life cycle of *Babesia* spp. requires complex interactions with its two hosts, which are the tick vector and the vertebrate host. The efficiency of tick acquisition and of transovarial and trans-stadial transmission of *B. bovis* and *B. bigemina* by *R. microplus*, involved in the long-term persistence of *Babesia* spp. in nature, is still poorly understood and warrants further investigations [28,29].

Interestingly, *A. marginale* was detected in *R. microplus* from both islands, but the infection rate reported in ticks from Guadeloupe seemed lower compared to in Martinique. The same trend had been reported during previous seroprevalence studies [7,8,10]. Anaplasmosis can be transmitted by vectors other than ticks, and some cattle breeds are known to be more susceptible than others to *Anaplasma* infection [10]. The difference in *Anaplasma* infection rate in ticks between the two islands may have been due to differences in the cattle populations. Indeed, there are mainly local Creole and mixed European-Creole breeds in Guadeloupe. These are known to be more resistant to anaplasmosis than Brahman and European breeds, which are the main breeds reared in Martinique [10]. In addition, other factors, including differences in the population dynamics of alternate vectors such as flies, may also have contributed to this difference.

Among the other known TBPs detected, we also found pathogens with low health impact in the Caribbean, almost considered as endosymbionts, such as *R. africae*, *T. velifera* and *T. mutans* in their *A. variegatum* vector and surprisingly in *R. microplus* ticks.

With almost all of the *A. variegatum* found to be infected, the *R. africae* infection rate was the highest ever reported in the Caribbean [9,13,14,30]. As *A. variegatum* is both the vector and the reservoir of the pathogen, with transovarial and trans-stadial transmission rates reaching 100%, this high level of *R. africae* infection is not surprising per se [14,31]. Interestingly, the high *R. africae* infection rate in vector ticks, associated with a very low number of African tick-bite fever cases in the Caribbean, highlights the difficulty, in some cases, of clearly distinguishing between endosymbiosis and pathogenicity [9,15]. The biological relationship between *R. africae* and *A. variegatum* as well as the strain variety and virulence of *R. africae* in the Caribbean should be investigated in order to better assess risks and guide prevention measures, especially for travelers [23,24,32]. The absence of direct identification of *R. africae* in *R. microplus* ticks was probably due to lower sensitivity of the specific target design compared to the genus target design. Indeed, *Rickettsia* spp.-positive *R. microplus* samples displayed rather high Ct values, suggesting a low infection level that may have been below the detection limit for *R. africae*. The unusual presence of *R. africae* in *R. microplus* ticks may have been due to the co-occurrence of the two tick species, *R. microplus* and *A. variegatum*, on cattle. As the ticks here were collected partially engorged, the presence of *R. africae* in *R. microplus* may have been due to bacteria circulating in cattle blood picked up by engorging ticks, or to cross-contamination with *R. microplus* ticks co-feeding next to infected *A. variegatum* [33,34].

This study provides the first update on the detection of *T. mutans* and *T. velifera* in Caribbean ticks. Indeed, references to these parasites in the Caribbean are relatively old, and no prevalence studies have been conducted since, whether in ticks or in cattle [5,6,35]. The low pathogenicity of these piroplasms may explain the lack of diagnoses and the scarcity of information available on their distribution and prevalence in the Caribbean. However, these parasite species may play an important role in theileriosis management and protection, as chronically infected cattle can develop immunity and heterologous protection against other pathogenic *Theileria* species, such as *Theileria parva* [36]. Unfortunately, these detections still require further investigations as we did not succeed in confirming these results by conventional or nested PCR, suggesting either a level of infection below the detection threshold, or simply false signals.

Lastly, the high-throughput microfluidic real-time PCR system enabled the detection of unexpected and/or potentially new microorganisms, leading to the recovery of nucleotide sequences of *Anaplasma* spp., *Ehrlichia* spp., *Borrelia* spp. and *Leishmania* spp. in ticks collected in Guadeloupe and Martinique.

The *Ehrlichia* sp. Tick428 sequence detected here formed a cluster with other uncharacterized Ehrlichia species detected in ticks from Asia and Africa [13,37,38,39,40,41]. However, given the highly conserved nature of the 16S rRNA genes, we could not more accurately define phylogenetic relationships within the *Ehrlichia* species group. The *Anaplasma* sp. Tick314 sequence was identified as *Candidatus* Anaplasma boleense, a bacterium described in ticks and mosquitoes in China [40,42]. No further information is available regarding the epidemiology of *Candidatus* Anaplasma boleense. These observations highlight the need to set up characterization studies. Indeed, high-throughput detection technologies can highlight the presence of DNA from potentially new microorganisms, but it will still be necessary to isolate and characterize them in order to first confirm their existence and then determine whether their presence in ticks poses a risk to public or animal health.

Here we provided the first report of *Borrelia* spp. in ticks from Guadeloupe and Martinique. Two different sequences were recovered, according to the tick species analyzed. In *A. variegatum*, a sequence named *Borrelia* sp. Tick7 was detected and was closely related to *B. anserina*, the agent of avian spirochetosis. Both of them seemed to define an intermediate position between the relapsing fever and Lyme disease groups. In contrast, the *Borrelia* sp. Tick457 sequence found in the *R. microplus* sample, clustered with uncharacterized *Borrelia* spp. described *R. microplus* specimens from Madagascar and Brazil, such as *Borrelia* sp. strain Mo063b and *Borrelia* sp. BR, and with relapsing fever Borrelia species encountered in hard ticks, including *Borrelia lonestari* and *B. theileri* [43,44]. Interestingly, the same observations had recently been made regarding *Borrelia* spp. found in *A. variegatum* and *R. microplus* ticks from Ethiopia and Côte d’Ivoire [45,46]. As *A. variegatum* and *R. microplus* were imported into the Caribbean from Africa during the time of the Atlantic triangular trade, we may have detected bacteria probably characterized by an old introduction through infected ticks and subsequent local evolution within their vector over a long period [4,47]. *Borrelia* spp. and borreliosis case reports in the Caribbean are scarce and still being debated. In Cuba, one study suggested the presence of antibodies to *Borrelia burgdorferi* sensu stricto in human sera associated with clinical cases of Lyme disease-like syndrome [48,49]. However, the real specificity of these serum antibodies has been questioned [50]. In the US Virgin Islands, seropositivity for *Borrelia hermsii* and closely related species was reported in association with a human case of relapsing fever [51]. Lastly, erythema migrans-like skin lesions and illness were reported in four Caribbean nationals [52]. Regarding the importance of *Borrelia* spp. for human and animal health, the characterization of these potential new *Borrelia* species that seemed associated with tropical tick species requires further investigation.

Lastly, *Leishmania* spp. were detected in *R. microplus* specimens from Martinique, and one sequence was identified as *Leishmania martiniquensis* Tick389 (accession number MK049850). Studies on *Leishmania* nomenclature have highlighted the fact that isolates of “*L. siamensis*” have never been officially characterized and that, therefore, this name should not be used [53,54,55,56]. Thus, since all the sequences, except one, reported as “*L. siamensis*” in databases should be considered as synonyms of *L. martiniquensis*, we assumed the occurrence of *L. martiniquensis* here. Parasites of the genus *Leishmania* are usually transmitted by female phlebotomine sand flies (Diptera: Psychodidae: Phlebotominae) and generally involve a wide variety of animal species, mainly including dogs and canids in the epidemiological cycle. They are responsible for leishmaniasis, a zoonosis widespread in tropical and subtropical areas [56]. *L. martiniquensis* belongs to the *L. enriettii* complex and has been described in Martinique and Thailand, where it was responsible for both cutaneous and visceral leishmaniosis [53,55,56,57,58]. *L. martiniquensis* is suspected to be endemic in Martinique [57]. Although phlebotomines and rodents are present in Martinique, neither vectors nor reservoirs of this parasite have yet been described [57]. Our study represents the first report of *L. martiniquensis* in *R. microplus* ticks from the French West Indies. Although *Leishmania* spp. have been reported in ticks (*L. infantum* in *R. sanguineus* s.l., and *L. guyanensis* in *R. microplus* ticks in Peru, for example), the role of ticks in *Leishmania* transmission is still being debated, and no evidence of vector capacity has been reported yet [59,60,61]. Moreover, the finding of *Leishmania* spp. in a tick species that feeds mainly on cattle also raises questions about the potential role of cattle in the epidemiology of leishmaniasis [62,63]. The participation of ticks in *Leishmania* epidemiology warrants further investigation, especially since *R. microplus* ticks could parasitize humans [64].

Surprisingly, co-infections with two or more TBPs were found in more than 50% of the infected ticks, both for *A. variegatum* and *R. microplus* and on the two islands. In addition, we could not identify any exclusion of infection between pathogens. These observations illustrate the efficiency of ticks as reservoirs of multiple pathogens with no apparent significant effects on their life traits.

To conclude, although screening tools are useful for the discovery of pathogens in ticks, the epidemiological significance of such results warrants further analysis. Detecting a microorganism’s DNA in ticks, especially in partially engorged ticks removed from the host, does not necessarily mean that the ticks are the biological vector of this microorganism; however, it provides useful information to supplement vector competence studies [16]. Nevertheless, the detection of potentially new microorganisms in ticks from the French West Indies has opened up new research perspectives for the future on the epidemiology of TBPs in the Caribbean. A region-wide epidemiological survey on TBPs in ticks collected in different countries and territories of the Caribbean area, organized in collaboration with the Caribbean Animal Health Network (CaribVET) in order to strengthen our results, may be an interesting way to supplement and strengthen some of this paper’s findings.

## 4. Materials and Methods 

### 4.1. Ticks Collected in Guadeloupe and Martinique

The ticks used in this study were collected as part of two separate epidemiological surveys conducted in Guadeloupe (between February 2014 and January 2015) and Martinique (between February and March 2015), respectively. In Guadeloupe, adult ticks (any species, any sex) were collected from 40 cattle originating from 22 different herds that were sampled in nine localities situated in six different biotopes (urban area, dry coastal regions, valleys and hills, evergreen seasonal forest, sub-mountainous rainforest and swamp forest). In Martinique, engorged females of *R. microplus* only were collected from cattle in 29 farms participating in a study on acaricide resistance in ticks. All the ticks were collected from cattle with the permission of farmers and cattle owners. The ticks were morphologically identified at species level [65]. A total of 578 adult ticks were included in the study: 132 *A. variegatum* and 165 *R. microplus* ticks from Guadeloupe and 281 *R. microplus* ticks from Martinique (see maps, Figure 2). The GPS coordinates of the tick collection sites are available in Appendix A. All the ticks were partially engorged and then stored at −80 °C.

### 4.2. DNA Extraction of Ticks Collected in Guadeloupe and Martinique

For 20 mg of tick, 1 mL of recently prepared PBS 1X was added to the sample. The ticks were then washed by gently shaking for 2–3 min at 7 Hz/s in a TissueLyser (Qiagen, Hilden, Germany). After discarding the supernatant, the ticks were frozen at −80 °C for 15–20 min. A steel ball was then added, and the samples were crushed twice for 2 min at 30 Hz/s with the TissueLyser (Qiagen, Hilden, Germany). A total of 450 µL of fresh PBS 1X was added to the samples. The samples were vortexed for 10 s and then centrifuged for 2–3 min at 8000× *g*. Lastly, 20 µL of Proteinase K was added to 150 µL of crushed tick sample, and DNA was extracted using the NucleoSpin® 96 Virus Core Kit (Macherey-Nagel, Düren, Germany) and the Biomek4000 automated platform (Beckman Coulter, Villepinte, France). This protocol enables the simultaneous extraction of both DNA and RNA. Total nucleic acid per sample was eluted in 160 µL of rehydration solution and stored at −80 °C until further use. All the *A. variegatum* ticks were individually extracted, and both individual and pooled extraction have been performed on *R. microplus* ticks. Indeed, as some *R. microplus* specimens were too small to be treated individually (20 mg of tick required), pools of two to four ticks have been carried out when required.

### 4.3. Assay Design

The list of pathogens to be monitored, the sets of primers and probes required for their detection, as well as the targeted genes are shown in Table 1. Some of the oligonucleotides were specifically designed for the purposes of this study; the others came from Michelet et al., 2014 [18]. The newly developed oligonucleotides were validated for a range of dilutions of positive controls, including cultures, plasmids and DNA samples (Table 1, Appendix A), by real-time TaqMan PCR assays on a LightCycler® 480 (LC480) (Roche Applied Science, Germany). More information on positive control origins are available in Appendix A. Real-time PCR assays were performed with LightCycler® 480 Probe Master Mix 1× (Roche Applied Science, Penzberg, Germany), using 200 nM of primers and probes in a final volume of 12 µL, and 2 µL of control DNA was added. The thermal cycling program was as follows: 95 °C for 5 min, 45 cycles at 95 °C for 10 s and 60 °C for 15 s, and one final cooling cycle at 40 °C for 10 s.

### 4.4. Pre-Amplification of DNA Samples

All the DNA samples were subject to pre-amplification in order to enrich the pathogenic DNA content compared with tick DNA. PerfeCTa® PreAmp SuperMix (Quanta Biosciences, Beverly, MA, USA) was used for DNA pre-amplification following the manufacturer’s instructions. All the primers were pooled (except those targeting the tick species), with a final and equal concentration of 45 nM each. The pre-amplification reaction was performed in a final volume of 5 µL containing 1 µL of PerfeCTa PreAmp SuperMix (5X), 1.25 µL of pooled primer mix, 1.25 µL of DNA and 1.5 µL of Milli-Q water, with one cycle at 95 °C for 2 min and 14 cycles at 95 °C for 10 s and 60 °C for 3 min. At the end of the cycling program, the reactions were 1:10 diluted. The pre-amplified DNA were stored at −20 °C until use.

### 4.5. High-Throughput Microfluidic Real-Time PCR

High-throughput microfluidic real-time PCR amplifications were performed using the BioMark™ real-time PCR system (Fluidigm, South San Francisco, CA, USA) and 96.96 dynamic arrays (Fluidigm, South San Francisco, CA, USA), enabling up to 9216 individual reactions to be performed in one run [18]. Real-time PCRs were performed using 6-carboxyfluorescein (6-FAM)- and Black Hole Quencher (BHQ1)-labeled TaqMan probes with TaqMan Gene Expression Master Mix (Applied Biosystems, Foster City, CA, USA) following the manufacturer’s instructions. The cycling conditions were as follows: 2 min at 50 °C and 10 min at 95 °C, followed by 40 cycles of two-step amplification for 15 s at 95 °C and 1 min at 60 °C. The BioMark™ real-time PCR system was used for data acquisition and the Fluidigm real-time PCR analysis software for Ct value determination. Three kinds of controls per chip were used for experiment validation: a negative water control to exclude contamination; a DNA extraction control, corresponding to primers and probes targeting a portion of the 16S rRNA gene of ticks; and an internal control, to check the presence of PCR inhibitors made of DNA from Escherichia coli strain EDL933, added to each sample with specific primers and probes targeting the eae gene [66]. For the relative specificity analysis of the newly designed Biomark system, DNA of 62 positive controls were used as template (Appendix A). Then, for the epidemiological survey of TBPs in Caribbean ticks, the 523 DNA samples of *A. variegatum* and *R. microplus* from Guadeloupe and Martinique were used as template.

### 4.6. Infection Rates for Ticks from the French West Indies

Depending on the tick species and the island of origin, for each detected pathogen, infection rates (the proportion of infected ticks divided by the total number of ticks analyzed) were calculated. The majority of the samples were single specimens of ticks. When ticks were too small to be treated individually, they were grouped into pools of two to four specimens. Thus, of the 523 samples analyzed, 47 consisted of a pool of two to four tick specimens. The final estimation of infection rates also includes the pools and is therefore expressed as the minimum (assuming at least one positive tick in the pools) and maximum (assuming all positive ticks in the pools) proportions of infected ticks out of the total number of ticks analyzed.

### 4.7. PCRs and Sequencing for the Confirmation of Results

Conventional PCRs/nested PCRs using primers targeting different genes or regions than those of the BioMark™ system were used to confirm the presence of pathogenic DNA in some field samples and positive controls (Table 3). PCR products were then sequenced by Sanger sequencing approach performed by Eurofins MWG Operon (BIOMNIS-EUROFINS GENOMICS, Nantes, France). Sequences obtained were assembled using BioEdit software (Ibis Biosciences, Carlsbad, CA, USA). An online BLAST (Basic Local Alignment Search Tool) search was used to compare the nucleotide sequences found in this study to reference sequences listed in GenBank sequence databases (NCBI).

### 4.8. Phylogenetic Sequence Analysis 

Alignments were performed using ClustalW [72]. Maximum likelihood trees were generated by 1000 bootstrap repetitions based on the Tamura-Nei model [73] in MEGA7 [74]. The initial tree(s) for the heuristic search were obtained automatically by applying neighbor-joining and BioNJ algorithms to a matrix of pairwise distances estimated using the maximum composite likelihood (MCL) approach and then selecting the topology with superior log likelihood value. The tree was drawn to scale, with branch lengths measured in the number of substitutions per site. The codon positions included were 1st+2nd+3rd+Noncoding. All positions containing gaps and missing data were eliminated. Further information is provided in the figure legends.

## 5. Conclusions

Our study demonstrated the high ability of microfluidic real-time PCR technology to provide a rapid overview of the diversity of TBPs of veterinary and medical importance present in ticks from the Caribbean. This innovative high-throughput tool is promising and could significantly improve the surveillance and exploration of TBPs, enabling the rapid screening of multiple microorganisms especially in regions where few epidemiological data are available and TBDs are numerous.

## Figures and Tables

**Figure 1 pathogens-09-00176-f001:**
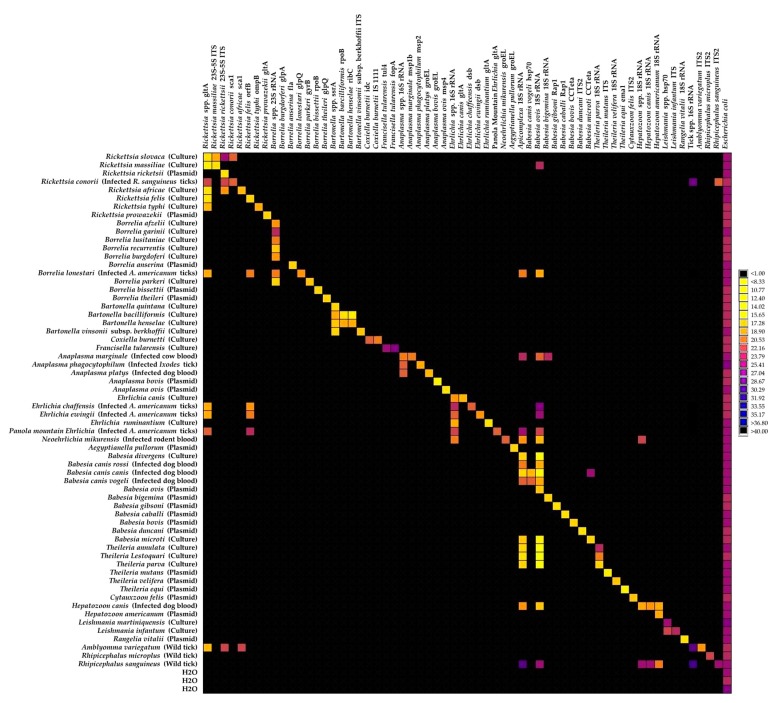
BioMark^TM^ dynamic array system specificity test (96.96 chip). Each square corresponds to a single real-time PCR reaction, where rows indicate the pathogen in the sample and columns represent the target of the primer/probe set. Ct values for each reaction are represented by a color gradient; the color scale is shown on the right y-axis. The darkest shades of blue and black squares are considered as negative reactions with Ct > 30.

**Figure 2 pathogens-09-00176-f002:**
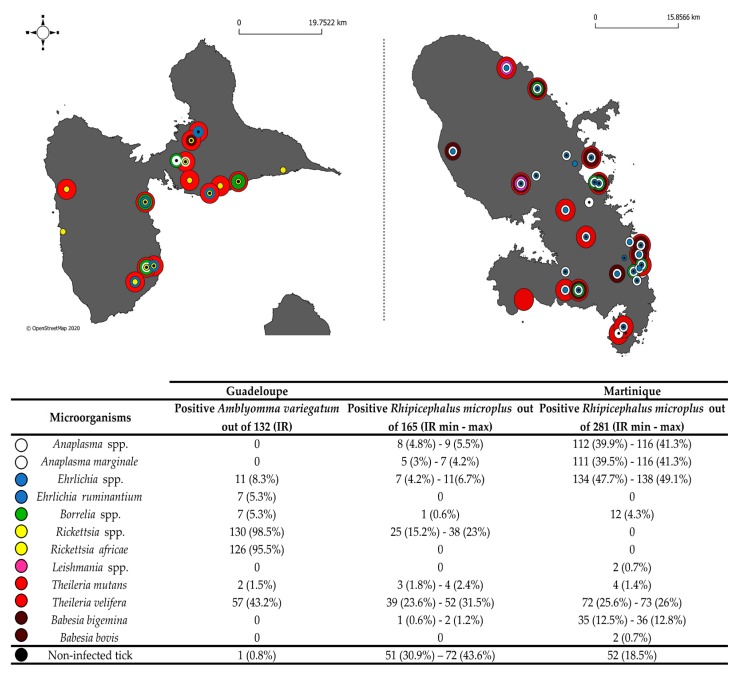
Infection rates in ticks collected in Guadeloupe and Martinique. Number of positive *A. variegatum* ticks (out of 132) and *R. microplus* ticks from Guadeloupe (out of 165) and Martinique (out of 281). On the maps, black dots indicate the collection sites of non-infected tick samples and colored dots indicate the collection sites of infected tick samples; The dot color determine the bacterial and parasitic genus of the microorganism found as indicated in the table; IR: Infection rate. As some *R. microplus* samples were pooled, we have presented minimum and maximum tick infection rates.

**Figure 3 pathogens-09-00176-f003:**
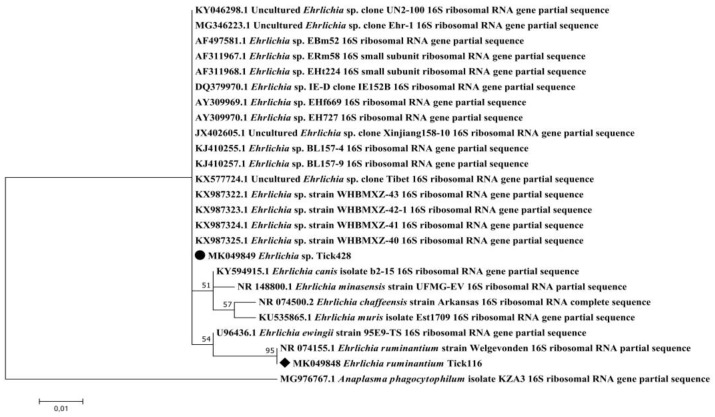
Phylogenetic analysis of 16S rRNA sequences of *Ehrlichia* spp. Phylogenetic analysis of 16S rRNA sequences of *Ehrlichia* spp. using the maximum likelihood method based on the Tamura–Nei model. In the phylogenetic tree, GenBank sequences, species designations and strain names are given. The sequences investigated in the present study are marked with a black circle (*Ehrlichia* sp. Tick428, accession number MK049849) and a black diamond (*Ehrlichia ruminantium* Tick116, accession number MK049848). The tree with the highest log likelihood (−413.76) is shown. The percentage of trees in which the associated taxa clustered together is shown above the branches (bootstrap values). The analysis involved 25 nucleotide sequences. There were a total of 206 positions in the final dataset.

**Figure 4 pathogens-09-00176-f004:**
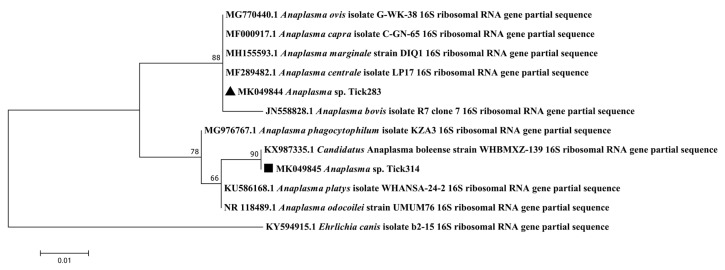
Phylogenetic analysis of 16S rRNA sequences of *Anaplasma* spp. Phylogenetic analysis of 16S rRNA sequences of *Anaplasma* spp. using the maximum likelihood method based on the Tamura–Nei model. In the phylogenetic tree, GenBank sequences, species designations and strain names are given. The sequences investigated in the present study are marked with a black triangle (*Anaplasma* sp. Tick283, accession number MK049844) and a black square (*Anaplasma* sp. Tick314, accession number MK049845). The tree with the highest log likelihood (−473.44) is shown. The percentage of trees in which the associated taxa clustered together is shown above the branches (bootstrap values). The analysis involved 12 nucleotide sequences. There were a total of 243 positions in the final dataset.

**Figure 5 pathogens-09-00176-f005:**
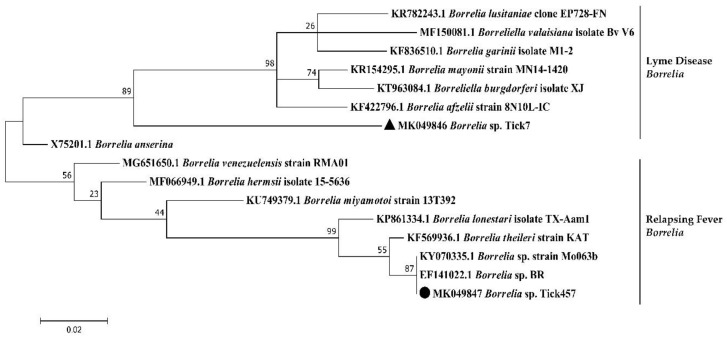
Phylogenetic analysis of flaB sequences of *Borrelia* spp. Phylogenetic analysis of flaB sequences of *Borrelia* spp. using the maximum likelihood method based on the Tamura–Nei model. In the phylogenetic tree, GenBank sequences, species designations and strain names are given. The sequences investigated in the present study are marked with a black circle (*Borrelia* sp. Tick457, accession number MK049847) and a black triangle (*Borrelia* sp. Tick7, accession number MK049846). The Lyme disease and relapsing fever clades of *Borrelia* are marked. The tree with the highest log likelihood (−963.24) is shown. The percentage of trees in which the associated taxa clustered together is shown above the branches (bootstrap values). The analysis involved 16 nucleotide sequences. There was a total of 245 positions in the final dataset.

**Figure 6 pathogens-09-00176-f006:**
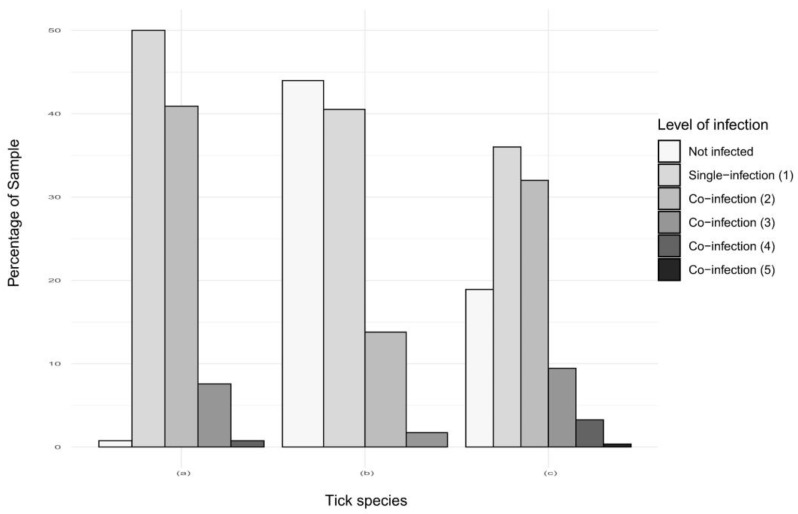
Co-infections detected in (a) *Amblyomma variegatum* (n = 132 samples) and (b) *Rhipicephalus microplus* collected in Guadeloupe (n = 116 samples) and (c) *Rhipicephalus microplus* collected in Martinique (n = 275 samples).

**Table 1 pathogens-09-00176-t001:** List of primer/probe sets constituting the BioMark system, with the positive controls used for their validation (new designs mainly). *: Design from Michelet et al., 2014 [18]. **: include all the controls belonging to the genus described in the table and targeted by specific design. Plasmids used as control are recombinant PBluescript IISK+ containing the target gene.

Microorganisms	Target	Design Name	Sequence (5′ à 3′)	Length (bp)	Controls
*Rickettsia* spp.	gltA	Rick_spp_gltA_F	GTCGCAAATGTTCACGGTACTT	78	**, Culture of *R. slovaca*
		Rick_spp_gltA_R	TCTTCGTGCATTTCTTTCCATTG		
		Rick_spp_gltA_P	TGCAATAGCAAGAACCGTAGGCTGGATG		
*Rickettsia massiliae **	ITS	Ri_ma_ITS_F	GTTATTGCATCACTAATGTTATACTG	128	Culture
		Ri_ma_ITS_R	GTTAATGTTGTTGCACGACTCAA		
		Ri_ma_ITS_P	TAGCCCCGCCACGATATCTAGCAAAAA		
*Rickettsia rickettsii **	ITS	Ri_ri_ITS_F	TCTACTCACAAAGTTATCAGGTTAA	124	Plasmid
		Ri_ri_ITS_R	CCTACGATACTCAGCAAAATAATTT		
		Ri_ri_ITS_P	TCGCTGGATATCGTTGCAGGACTACAG		
*Rickettsia conorii*	sca1	Ri_co_sca1_F	GTAGATGCTTCATAGAATACTGC	88	Infected *Rhipicephalus sanguineus* s.l.
		Ri_co_sca1_R	CCAAATTTAGTCTACCTTGTGATC		
		Ri_co_sca1_P	TCCTCCTGACGTATTAAAAGAAGCTGAAGCT		
*Rickettsia africae*	sca1	Ri_af_sca1_F	GATACGACAAGTACCTCGCAG	122	Culture
		Ri_af_sca1_R	GGATTATATACTTTAGGTTCGTTAG		
		Ri_af_sca1_P	CAGATAGGAACAGTAATTGTAACGGAACCAG		
*Rickettsia felis*	orfB	Ri_fel_orfB_F	ACCCTTTTCGTAACGCTTTGC	163	Culture
		Ri_fel_orfB_R	TATACTTAATGCTGGGCTAAACC		
		Ri_fel_orfB_P	AGGGAAACCTGGACTCCATATTCAAAAGAG		
*Rickettsia typhi*	ompB	Ri_typ_ompB_F	CAGGTCATGGTATTACTGCTCA	133	Culture
		Ri_typ_ompB_R	GCAGCAGTAAAGTCTATTGATCC		
		Ri_typ_ompB_P	ACAAGCTGCTACTACAAAAAGTGCTCAAAATG		
*Rickettsia prowazekii*	gltA	Ri_pro_gltA_F	CAAGTATCGGTAAAGATGTAATCG	151	Plasmid
		Ri_pro_gltA_R	TATCCTCGATACCATAATATGCC		
		Ri_pro_gltA_P	ATATAAGTAGGGTATCTGCGGAAGCCGAT		
*Borrelia* spp. ***	23S rRNA	Bo_bu_sl_23S_F	GAGTCTTAAAAGGGCGATTTAGT	73	**, Culture of *B. afzelii, B. garinii*, *B. valaisiana*, *B. lusitaniae*, *B. recurrentis*
		Bo_bu_sl-23S_R	CTTCAGCCTGGCCATAAATAG		
		Bo_bu_sl_23S_P	TAGATGTGGTAGACCCGAAGCCGAGT		
*Borrelia burgdorferi* sensu stricto	glpA	Bo_bu_glpA_F	GCAATTACAAGGGGGTATAAAGC	206	Culture
		Bo_bu_glpA_R	GGCGTGATAAGTGCACATTCG		
		Bo_bu_glpA_P	TTAATTAAACGGGGTGCATTCTTCTCAAGAATG		
*Borrelia anserina*	fla	Bor_ans_fla_F	GGAGCACAACAAGAGGGAG	76	Plasmid
		Bor_ans_fla_R	TTGGAGAATTAACCCCACCTG		
		Bor_ans_fla_P	TGCAAGCAACTCCAGCTCCAGTAGCT		
*Borrelia lonestari*	glpQ	Bor_lon_glpQ_F	GATCCAGAACTTGATACAACCAC	99	Infected *Amblyomma americanum*
		Bor_lon_glpQ_R	TTCATCTAGTGAGAAGTCAGTAG		
		Bor_lon_glpQ_P	AGTAATATCGTCCGTCTTCCCTAGCTCG		
*Borrelia parkeri*	gyrB	Bor_par_gyrB_F	GCAAAACGATTCAAAGTGAGTCC	184	Culture
		Bor_par_gyrB_R	CTCATTGCCTTTAAGAAACCACTT		
		Bor_par_gyrB_P	TTAAAACCAGCAACATGAGTTCCTCCTTCTC		
*Borrelia bissettii **	rpoB	Bo_bi_rpoB_F	GCAACCAGTCAGCTTTCACAG	118	Plasmid
		Bo_bi_rpoB_R	CAAATCCTGCCCTATCCCTTG		
		Bo_bi_rpoB_P	AAAGTCCTCCCGGCCCAAGAGCATTAA		
*Borrelia theileri*	glpQ	Bo_th_glpQ_F	GTGCTAACAAAGGACAATATTCC	213	Plasmid
		Bo_th_glpQ_R	GGTTAGTGGAAAACGGTTAGGAT		
		Bo_th_glpQ_P	TATTATAATTCACGAGCCAGAGCTTGACAC		
*Bartonella* spp.	ssrA	Bart_spp_ssrA_F	CGTTATCGGGCTAAATGAGTAG	118	**, Culture of *B. quintana*
		Bart_spp_ssrA_R	ACCCCGCTTAAACCTGCGA		
		Bart_spp_ssrA_P	TTGCAAATGACAACTATGCGGAAGCACGTC		
*Bartonella barcilliformis **	rpoB	Ba_ba_rpoB_F	GAAGAGTTTGTAGTTTGTCGTCA	105	Culture
		Ba_ba_rpoB_R	AGCAGCTACAGAAACCAACTG		
		Ba_ba_rpoB_P	TGCAGGTGAAGTTTTGATGGTGCCACG		
*Bartonella henselae*	ribC	Bar_he_ribC_F	GGGATGCGATTTAATAGTTCTAC	116	Culture
		Bar_he_ribC_R	CGCTTGTTGTTTTGATCCTCG		
		Bar_he_ribC_P	ACGTTATAGTAGCGAAAACTTAGAAATTGGTGC		
*Bartonella vinsonii* subsp. *berkhoffii*	ITS	Bar_vin_ITS_F	GGAATTGCTTAACCCACTGTTG	141	Culture
		Bar_vin_ITS_R	CCTTATTGATTTAGATCTGATGGG		
		Bar_vin_ITS_P2	AGAAACTCCCGCCTTTATGAGAGAAATCTCT		
*Coxiella burnetii* and *Coxiella*-like *	Icd	Co_bu_icd_F	AGGCCCGTCCGTTATTTTACG	74	Culture
		Co_bu_icd_R	CGGAAAATCACCATATTCACCTT		
		Co_bu_icd_P	TTCAGGCGTTTTGACCGGGCTTGGC		
	IS1111	Co_bu_IS111_F	TGGAGGAGCGAACCATTGGT	86	Culture
		Co_bu_IS111_R	CATACGGTTTGACGTGCTGC		
		Co_bu_IS111_P	ATCGGACGTTTATGGGGATGGGTATCC		
*Francisella tularensis* and *Francisella*-like *endosymbionts* *	tul4	Fr_tu_tul4_F	ACCCACAAGGAAGTGTAAGATTA	76	Culture
		Fr_tu_tul4_R	GTAATTGGGAAGCTTGTATCATG		
		Fr_tu_tul4_P	AATGGCAGGCTCCAGAAGGTTCTAAGT		
	fopA	Fr_tu_fopA_F	GGCAAATCTAGCAGGTCAAGC	91	Culture
		Fr_tu_fopA_R	CAACACTTGCTTGAACATTTCTAG		
		Fr_tu_fopA_P	AACAGGTGCTTGGGATGTGGGTGGTG		
*Anaplasma* spp.	16S rRNA	Ana_spp_16S_F	CTTAGGGTTGTAAAACTCTTTCAG	160	**
		Ana_spp_16S_R	CTTTAACTTACCAAACCGCCTAC		
		Ana_spp_16S_P	ATGCCCTTTACGCCCAATAATTCCGAACA		
*Anaplasma marginale **	msp1b	An_ma_msp1_F	CAGGCTTCAAGCGTACAGTG	85	Experimentally infected bovine blood sample
		An_ma_msp1_R	GATATCTGTGCCTGGCCTTC		
		An_ma_msp1_P	ATGAAAGCCTGGAGATGTTAGACCGAG		
*Anaplasma phagocytophilum **	msp2	An_ph_msp2_F	GCTATGGAAGGCAGTGTTGG	77	Infected *Ixodes* spp. tick
		An_ph_msp2_R	GTCTTGAAGCGCTCGTAACC		
		An_ph_msp2_P	AATCTCAAGCTCAACCCTGGCACCAC		
*Anaplasma platys **	groEL	An_pla_groEL_F	TTCTGCCGATCCTTGAAAACG	75	Infected canine blood sample
		An_pla_groEL_R	CTTCTCCTTCTACATCCTCAG		
		An_pla_groEL_P	TTGCTAGATCCGGCAGGCCTCTGC		
*Anaplasma bovis **	groEL	An_bo_groEL_F	GGGAGATAGTACACATCCTTG	73	Plasmid
		An_bo_groEL_R	CTGATAGCTACAGTTAAGCCC		
		An_bo_groEL_P	AGGTGCTGTTGGATGTACTGCTGGACC		
*Anaplasma ovis **	msp4	An_ov_msp4_F	TCATTCGACATGCGTGAGTCA	92	Plasmid
		An_ov_msp4_r	TTTGCTGGCGCACTCACATC		
		An_ov_msp4_P	AGCAGAGAGACCTCGTATGTTAGAGGC		
*Ehrlichia* spp. ***	16S rRNA	Neo_mik_16S_F	GCAACGCGAAAAACCTTACCA	98	**
		Neo_mik_16S_R	AGCCATGCAGCACCTGTGT		
		Neo_mik_16S_P	AAGGTCCAGCCAAACTGACTCTTCCG		
*Ehrlichia canis*	gltA	Eh_ca_gltA_F	GACCAAGCAGTTGATAAAGATGG	136	Culture
		Eh_ca_gltA_R	CACTATAAGACAATCCATGATTAGG		
		Eh_ca_gltA_P	ATTAAAACATCCTAAGATAGCAGTGGCTAAGG		
*Ehrlichia chaffeensis **	dsb	Eh_ch_dsb_F	TATTGCTAATTACCCTCAAAAAGTC	117	Infected *Amblyomma americanum*
		Eh_ch_dsb_R	GAGCTATCCTCAAGTTCAGATTT		
		Eh_ch_dsb_P	ATTGACCTCCTAACTAGAGGGCAAGCA		
*Ehrlichia ewingii **	dsb	Eh_ew_dsb_F	CAATACTTGGAGAAGCATCATTG	111	Infected *Amblyomma americanum*
		Eh_ew_dsb_R	TTGCTTATGGCTTAATGCTGCAT		
		Eh_ew_dsb_P	AAAGCAGTACGTGCAGCATTGGCTGTA		
*Ehrlichia ruminantium*	gltA	Eh_ru_gltA_F	CCAGAAAACTGATGGTGAGTTAG	116	Culture
		Eh_ru_gltA_R	AGCCTACATCAGCTTGAATGAAG		
		Eh_ru_gltA_P	AGTGTAAACTTGCTGTTGCTAAGGTAGCATG		
Panola Mountain *Ehrlichia*	gltA	Eh_PME_gltA_F	GCTAGTTATGAGTTAGAATGTAAAC	121	Infected *Amblyomma americanum*
		Eh_PME_gltA_R	TACTATAGGATAATCTTGAATCAGC		
		Eh_PME_gltA_P	TTGCTATCGCTAAAATTCCAAGTATGATTGCG		
*Neoehrlichia mikurensis **	groEL	Neo_mik_groEL_F	AGAGACATCATTCGCATTTTGGA	96	Infected rodent blood sample
		Neo_mik_groEL_R	TTCCGGTGTACCATAAGGCTT		
		Neo_mik_groEL_P	AGATGCTGTTGGATGTACTGCTGGACC		
*Aegyptianella pullorum*	groEL	Ae_pul_groEL_F	AGCCAGTATTATCGCTCAAGG	168	Plasmid
		Ae_pul_groEL_R	GCCTCACGTGCCTTCATAAC		
		Ae_pul_groEL_P	TGCTTCTCAGTGTAACGACAGGGTTGG		
*Apicomplexa*	18S rRNA	Apic_18S_F	TGAACGAGGAATGCCTAGTATG	104	**, Infected canine blood sample, with *B. canis rossi, B. canis canis*; Culture of *B. divergens, T. lestoquari, T. annulata*
		Apic_18S_R	CACCGGATCACTCGATCGG		
		Apic_18S_S	TAGGAGCGACGGGCGGTGTGTAC		
*Babesia canis vogeli **	hsp70	Ba_vo_hsp70_F	TCACTGTGCCTGCGTACTTC	87	Infected canine blood sample
		Ba_vo_hsp70_R	TGATACGCATGACGTTGAGAC		
		Ba_vo_hsp70_P	AACGACTCCCAGCGCCAGGCCAC		
*Babesia ovis **	18S rRNA	Ba_ov_18S_F	TCTGTGATGCCCTTAGATGTC	92	Plasmid
		Ba_ov_18S_R	GCTGGTTACCCGCGCCTT		
		Ba_ov_18S_P	TCGGAGCGGGGTCAACTCGATGCAT		
*Babesia bigemina **	18S rRNA	Ba_big_RNA18S_F	ATTCCGTTAACGAACGAGACC	99	Plasmid
		Ba_big_RNA18S_R	TTCCCCCACGCTTGAAGCA		
		Ba_big_RNA18S_P	CAGGAGTCCCTCTAAGAAGCAAACGAG		
*Babesia gibsoni*	Rap1	Ba_gib_rap1_F	CTCTTGCTCATCATCTTTTCGG	130	Plasmid
		Ba_gib_rap1_R	TCAGCGTATCCATCCATTATATG		
		Ba_gib_rap1_S	TTTAATGCGTGCTACGTTGTACTTCCCAAAG		
*Babesia caballi **	Rap1	Ba_cab_rap1_F	GTTGTTCGGCTGGGGCATC	94	Plasmid
		Ba_cab_rap1_R	CAGGCGACTGACGCTGTGT		
		Ba_cab_rap1_P	TCTGTCCCGATGTCAAGGGGCAGGT		
*Babesia bovis **	CCTeta	Ba_bo_CCTeta_F	GCCAAGTAGTGGTAGACTGTA	100	Plasmid
		Ba_bo_CCTeta_R	GCTCCGTCATTGGTTATGGTA		
		Ba_bo_CCTeta_P	TAAAGACAACACTGGGTCCGCGTGG		
*Babesia duncani **	ITS2	Ba_du_ITS_F	ATTTCCGTTTGCGAGAGTTGC	87	Plasmid
		Ba_du_ITS_R	AGGAAGCATCAAGTCATAACAAC		
		Ba_du_ITS_P	AACAAGAGGCCCCGAGATCAAGGCAA		
*Babesia microti **	CCTeta	Bab_mi_CCTeta_F	ACAATGGATTTTCCCCAGCAAAA	145	Culture
		Bab_mi_CCTeta_R	GCGACATTTCGGCAACTTATATA		
		Bab_mi_CCTeta_P	TACTCTGGTGCAATGAGCGTATGGGTA		
*Theileria parva **	18S rRNA	Th_pa_18S_F	GAGTATCAATTGGAGGGCAAG	173	Culture
		Th_pa_18S_R	CAGACAAAGCGAACTCCGTC		
		Th_pa_18S_P	AAATAAGCCACATGCAGAGACCCCGAA		
*Theileria mutans*	ITS	The_mu_ITS_F	CCTTATTAGGGGCTACCGTG	119	Plasmid
		The_mu_ITS_R	GTTTCAAATTTGAAGTAACCAAGTG		
		The_mu_ITS_P	ATCCGTGAAAAACGTGCCAAACTGGTTAC		
*Theileria velifera*	18S rRNA	The_ve_18S_F	TGTGGCTTATCTGGGTTCGC	151	Plasmid
		The_ve_18S_R	CCATTACTTTGGTACCTAAAACC		
		The_ve_18S_P	TTGCGTTCCCGGTGTTTTACTTTGAGAAAG		
*Theileria equi*	ema1	Th_eq_ema1_F4	CGGCAAGAAGCACACCTTC	167	Plasmid
		Th_eq_ema1_R4	TGCCATCGCCCTTGTAGAG		
		Th_eq_ema1_P4	AAGGCTCCAGGCAAGCGCGTCCT		
*Cytauxzoon felis*	ITS2	Cy_fel_ITS2_F	AAGATCCGAACGGAGTGAGG	119	Plasmid
		Cy_fel_ITS2_R	GTAGTCTCACCCAATTTCAGG		
		Cy_fel_ITS2_S	AAGTGTGGGATGTACCGACGTGTGAG		
*Hepatozoon* spp.	18S rRNA	Hepa_spp_18S_F	ATTGGCTTACCGTGGCAGTG	175	**
		Hepa_spp_18S_R	AAAGCATTTTAACTGCCTTGTATTG		
		Hepa_spp_18S_S	ACGGTTAACGGGGGATTAGGGTTCGAT		
*Hepatozoon canis*	18S rRNA	He_can_18S_F	TTCTAACAGTTTGAGAGAGGTAG	221	Infected canine blood sample
		He_can_18S_R	AGCAGACCGGTTACTTTTAGC		
		He_can_18S_S	AGAACTTCAACTACGAGCTTTTTAACTGCAAC		
*Hepatozoon americanum*	18S rRNA	He_ame_18S_F2	GGTATCATTTTGGTGTGTTTTTAAC	159	Plasmid
		He_ame_18S_R2	CTTATTATTCCATGCTCCAGTATTC		
		He_ame_18S_P2	AAAAGCGTAAAAGCCTGCTAAAAACACTCTAC		
*Leishmania* spp.	hsp70	Leish_spp_hsp70_F	CGACCTGTTCCGCAGCAC	78	** and culture of *L. martiniquensis*
		Leish_spp_hsp70_R	TCGTGCACGGAGCGCTTG		
		Leish_spp_hsp70_S	TCCATCTTCGCGTCCTGCAGCACG		
*Leishmania infantum*	ITS	Le_inf_ITS_F	CGCACCGCCTATACAAAAGC	103	Culture
		Le_inf_ITS_R	GTTATGTGAGCCGTTATCCAC		
		Le_inf_ITS_S	ACACGCACCCACCCCGCCAAAAAC		
*Rangelia vitalii*	18S rRNA	Ra_vit_18S_F	TAACCGTGCTAATTGTAGGGC	92	Plasmid
		Ra_vit_18S_R	GAATCACCAAACCAAATGGAGG		
		Ra_vit_18S_S	TAATACACGTTCGAGGGCGCGTTTTGC		
*Tick* spp.	16S rRNA	Tick_spp_16S_F	AAATACTCTAGGGATAACAGCGT	99	**
		Tick_spp_16S_R	TCTTCATCAAACAAGTATCCTAATC		
		Tick_spp_16S_P	CAACATCGAGGTCGCAAACCATTTTGTCTA		
*Amblyomma variegatum*	ITS2	Amb_var_ITS2_F	GCCAGCCTCTGAAGTGACG	117	Tick extract (Guadeloupe)
		Amb_var_ITS2_R	TTCTGCGGTTTAAGCGACGC		
		Amb_var_ITS2_P	TCTTGCCACTCGACCCGTGCCTC		
*Rhipicephalus microplus*	ITS2	Rhi_mic_ITS2_F	GCTTAAGGCGTTCTCGTCG	144	Tick extract (Galapagos Islands)
		Rhi_mic_ITS2_R	CAAGGGCAGCCACGCAG		
		Rhi_mic_ITS2_P	TAGTCCGCCGTCGGTCTAAGTGCTTC		
*Rhipicephalus sanguineus* sensu lato	ITS2	Rhi_san_ITS2_F	TTGAACGCTACGGCAAAGCG	110	Tick extract (France)
		Rhi_san_ITS2_R	CCATCACCTCGGTGCAGTC		
		Rhi_san_ITS2_P	ACAAGGGCCGCTCGAAAGGCGAGA		

**Table 2 pathogens-09-00176-t002:** Homology between the deposited sequences and reference sequences in GenBank (T: Sample number tested by conventional assay; S: Sample number which allowed sequence recovery; AN: Accession number of the recovered sequence; L: recovered sequence length (bp); Id%: percentage of nucleotide identity between recovered and reference sequences).

Biomark Id	Sequence Name	T	S	AN	L	Closest Homology	Id%	Reference
*Rickettsia* spp.	*Rickettsia africae* Tick208	30	14	MK049851	248	*Rickettsia africae*	100	AF123706.1
*Leishmania* spp.	*Leishmania martiniquensis* Tick389	2	1	MK049850	272	*Leishmania martiniquensis*	100	AF303938.1
						*Leishmania siamensis*	100	GQ226033.1
*Borrelia* spp.	*Borrelia* sp. Tick7	30	1	MK049846	245	*Borrelia anserina*	90	X75201.1
	*Borrelia* sp. Tick457		4	MK049847	327	*Borrelia* sp. BR	100	EF141022.1
						*Borrelia* sp. strain Mo063b-flaB	100	KY070335.1
						*Borrelia theileri*	99	KF569936.1
*Anaplasma* spp.	*Anaplasma* sp. Tick314	2	2	MK049845	245	*Candidatus* Anaplasma boleense	100	KX987335.1
*Anaplasma marginale*	*Anaplasma* sp. Tick283	2	2	MK049844	244	*Anaplasma marginale*	100	MH155593.1
						*Anaplasma centrale*	100	MF289482.1
						*Anaplasma ovis*	100	MG770440.1
						*Anaplasma capra*	100	MF000917.1
						*Anaplasma phagocytophilum*	100	DQ648489.1
*Ehrlichia* spp.	*Ehrlichia* sp. Tick428	2	2	MK049849	246	*Ehrlichia* spp.	100	KY594915.1
						*Ehrlichia canis*	99	KY594915.1
						*Ehrlichia ewingii*	99	U96436.1
						*Ehrlichia chaffeensis*	99	NR_074500.2
						*Ehrlichia muris*	99	KU535865.1
						*Ehrlichia minasensis*	99	NR_148800.1
*Ehrlichia ruminantium*	*Ehrlichia ruminantium* Tick116	1	1	MK049848	207	*Ehrlichia ruminantium*	100	NR_074155.1
*Babesia bigemina*	*Babesia bigemina* Tick222	2	1	MK071738	99	*Babesia bigemina*	100	KP710227.1
*Babesia bovis*	*Babesia bovis* Tick497	2	2	MK071739	100	*Babesia bovis*	99	AB367921.1

**Table 3 pathogens-09-00176-t003:** Primers used to confirm the presence of pathogenic DNA in tick samples.

Pathogen	Targeted Gene	Primer Name	Sequence (5′ → 3′)	Length (bp)	References
*Rickettsia* spp.	gltA	Rsfg877	GGGGGCCTGCTCACGGCGG	381	[67]
		Rsfg1258	ATTGCAAAAAGTACAGTGAACA		
	ompB	Rc.rompB.4362p	GTCAGCGTTACTTCTTCGATGC	475	[68]
		Rc.rompB.4836n	CCGTACTCCATCTTAGCATCAG		
		Rc.rompB.4496p	CCAATGGCAGGACTTAGCTACT	267	
		Rc.rompB.4762n	AGGCTGGCTGATACACGGAGTAA		
*Anaplasma*/*Ehrlichia* spp.	16S rRNA	EHR16SD	GGTACCYACAGAAGAAGTCC	345	[69]
		EHR16SR	TAGCACTCATCGTTTACAGC		
*Borrelia* spp.	flaB	FlaB280F	GCAGTTCARTCAGGTAACGG	645	[70]
		FlaRL	GCAATCATAGCCATTGCAGATTGT		
		FlaB737F	GCATCAACTGTRGTTGTAACATTAACAGG	407	
		FlaLL	ACATATTCAGATGCAGACAGAGGT		
*Leishmania* spp.	SSU rRNA	R221	GGTTCCTTTCCTGATTTACG	603	[71]
		R332	GGCCGGTAAAGGCCGAATAG		
		R223	TCCATCGCAACCTCGGTT	358	
		R333	AAAGCGGGCGCGGTGCTG

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
