# Peer review of "Upscaling the Surveillance of Tick-Borne Pathogens in the French Caribbean Islands"

_pathogens, 2020, doi:10.3390/pathogens9030176_

Round 1

Reviewer 1 Report

This manuscript details an exploratory epidemiological study investigating the presence of a multitude of microbes in ticks collected from cattle in the French Caribbean islands.  The study employed a high-throughput microfluidic qPCR system to provide and overview of the genera/species present in the collected ticks.  The study is of importance for detailing tick-borne microbes/pathogens in an understudied area of the world.

Introduction: 

  • R. sanguineus assay development is included in the results section.  However, there is not description of R. sanguineus in the the Introduction.  Since this tick is found in the Carribean and was part of assay development, it should be at least briefly mentioned/described in the Introduction.

Table 1 and 3:

  • Spacing needs revised to improve the layout of the table.

Table 2:

  • Needs reorganized.  What do the abbreviations stand for?  Clarify.

Figure 6:

  • Remove title.
  • (a), (b), (c) need clarified (modify legend appropriately).

Results:

  • Were all cell cultures, positive controls, etc. produced in-house?  A description or possibly a table listing the seed material origination, plasmid constructions, tick origins for all positive controls should be included with the manuscript.
  • Why were pools of ticks only used in some instances?
  • L166: “some of the results” - how many?  A distinct number should be provided.
  • L171-173:  Maps showing the locations of infected ticks would greatly enhance the results.
  • Why was 16S chosen for sequencing purposes?  It is the lease specific for MLST purposes?  
  • How many genes were attempted for failed sequencing?  Sequencing is key to validation.

Discussion:

  • L341:  What does the abbreviation POSEIDOM stand for?
  • It should be mentioned early on in the Discussion that this study only tested ticks collected from cattle (not from the environment) - which is a limitation in broad data interpretation.

Materials & Methods:

  • Section 4.3 - descriptions of the production/source of positive controls should be detailed.
  • Section 4.7 - descriptions of how the samples were prepared for sequencing should be included.

Appendix B:

- The use of the term pathogen should be re-considered, as not all tick-borne microbes are pathogens of humans/animals.

Author Response

We thanks the reviewer for his valuable comments. We tried to follow all the recommendations and performed the corrections as suggested.

Introduction: 

  • R. sanguineus assay development is included in the results section.  However, there is not description of R. sanguineus in the the Introduction.  Since this tick is found in the Carribean and was part of assay development, it should be at least briefly mentioned/described in the Introduction.
  • Addition of a brief mention of R. sanguineus sensu lato in the introduction, Line 47 and 49

Table 1 and 3:

  • Spacing needs revised to improve the layout of the table.
  • Spacing revised as requested

Table 2:

  • Needs reorganized.  What do the abbreviations stand for?  Clarify.
  • Spacing revised, meaning of the abbreviations have been added accordingly “Homology between the deposited sequences and reference sequences in GenBank (T: Sample number tested by conventional assay; S: Sample number which allowed sequence recovery; AN: Accession number of the recovered sequence; L: recovered sequence length (bp); Id%: percentage of nucleotide identity between recovered and reference sequences).”

Figure 6:

  • Remove title.
  • (a), (b), (c) need clarified (modify legend appropriately).
  • Title has been removed, and legend clarified accordingly “Co-infections detected in (a) Amblyomma variegatum (n=132 samples) and (b) Rhipicephalus microplus collected in Guadeloupe (n=116 samples) and (c) Rhipicephalus microplus collected in Martinique (n=275 samples)”.

Results:

  • Were all cell cultures, positive controls, etc. produced in-house?  A description or possibly a table listing the seed material origination, plasmid constructions, tick origins for all positive controls should be included with the manuscript.
  • Those information are available in supplementary data table S2.

  • Why were pools of ticks only used in some instances?
  • The methodology used for tick crushing and nucleic acid extraction required at least 20 mg of tick, but as some adult specimens of Rhipicephalus microplus collected were too small they have been pooled in order to reach this weight limit. Clarifications added Lines 539-542: “All the A. variegatum ticks were individually extracted, and. both individual and pooled extraction have been performed on R. microplus ticks. Indeed, as some R. microplus specimens were too small to be treated individually (20 mg of tick required), pools of two to four ticks have been carried out when required.”

  • L166: “some of the results” - how many?  A distinct number should be provided.
  • “some” has been removed since we tried to confirm at least one positive sample per microorganism detected with the BioMarkTM. The number of tested samples for validation is notified in Table 2.

  • L171-173:  Maps showing the locations of infected ticks would greatly enhance the results.
  • Maps in Figure 2 have been modified in order to present the location of non-infected and infected tick samples, as well as the type of infection reported, as suggested.

  • Why was 16S chosen for sequencing purposes?  It is the lease specific for MLST purposes?  
  • In order to validate the specificity of the detection results obtained with the BiomarkTM, we looked in the literature for already published conventional assays allowing to obtain amplicon length suitable for Sanger sequencing, and targeting gene well represented in sequence database. Among the assay selected, different target genes, not only the 16S rRNA genes, were used according to the microbial genus targeted (Table 3).
  • How many genes were attempted for failed sequencing?  Sequencing is key to validation.
  • Failed sequencing results were obtained only for the Theileria species, Theileria mutans and Theileria velifera. Only conventional and nested PCR targeting 18s rRNA genes were tested to confirm the presence of Theileria spp. The main limitation here regarding the use of another gene target was due to the lack of reference sequences of Theileria species in public Database.

Discussion:

  • L341:  What does the abbreviation POSEIDOM stand for?
  • POSEIDOM “Programme d’Options Spécifiques dus à l’Eloignement et à l’Insularité des DOM” (FR) can be translated by “Specific Options Program due to the remoteness and insularity of the overseas departments”. The meaning of the abbreviation has been added Line 350-352.

  • It should be mentioned early on in the Discussion that this study only tested ticks collected from cattle (not from the environment) - which is a limitation in broad data interpretation.
  • The information has been added accordingly lines 329-330 “We used the newly developed high-throughput microfluidic real-time PCR system to perform an exploratory epidemiological study of on TBPs and microorganisms potentially circulating in A. variegatum and R. microplus ticks collected on cattle in Guadeloupe and Martinique.”

Materials & Methods:

  • Section 4.3 - descriptions of the production/source of positive controls should be detailed.
  • Those information are available in supplementary data table S2. Reference to Table S2 added in this section, Line 505-506 “More information on positive controls origins are available in Table S2.”

  • Section 4.7 - descriptions of how the samples were prepared for sequencing should be included.
  • The sequencing of the amplicon have been outsourced, we sent our PCR products for Sanger sequencing by Eurofins MWG Operon (BIOMNIS-EUROFINS GENOMICS, France). Sequencing type added as requested, Line 551-552 “PCR products were then sequenced by Sanger sequencing approach performed by Eurofins MWG Operon (BIOMNIS-EUROFINS GENOMICS, France). Sequences obtained were assembled using BioEdit software (Ibis Biosciences, Carlsbad, CA, USA).”

Appendix B:

  • The use of the term pathogen should be re-considered, as not all tick-borne microbes are pathogens of humans/animals.
  • “pathogen” has been replaced by “microorganism” accordingly.

Reviewer 2 Report

        The study demonstrated the application of a high throughput microfluidic real-time PCR technology on the detection of tick-borne pathogens in the French Caribbean islands. Not only the known pathogens were successfully detected, but novel microorganisms were also discovered. The manuscript is generally well-written, however, some inconsistencies need to be clarified or modified before it could be considered for publication.

-p3, Table 1: Were all these primers and probes designed by the authors in the study? Or any of them were adopted from previous publications?

-p11, L172: B. ovis? or B. bovis? According to the description (p10, L144-145), Borrelia burgdorferi sensu stricto, Babesia ovis, and Rickettsia rickettsii were removed from the high throughput screening in the study because of the failure in detecting positive control or low specificity.

-p11, L175-176: Rickettsia spp. detected in 15.7-23.5% of the R. micropolus samples...Figure 2 shows 15.2-23%.

-p12, L189: 3.6-4.8%? Figure 2 shows 3-4.2%.

-p12, L201: 42.3%? Figure 2 shows 43.3%. 24.1-31.9%? Figure 2 shows 23.6-31.5%.

-p12, L211: 4.2-6.6%? Figure 2 shows 4.2-6.7%.

-p13, L227: Please provide the number of the ticks in the parenthesis, e.g. N/n.

-p14, L246: 5.1% or 5.3% (in Figure 2)?

Other comments:

-p2, L67: However, very few information are available...Information is an uncountable noun.

-p3, L104: sensu lato. The Latin referent words and phrases are usually italicized. This also happened in p3 Table 1 (Infected Rhipicephalus sanguineus s. l.), p4 (Borrelia burgdorferi sensu stricto), p8 (Rhipicephalus sanguineus s. l.), p8 L113, p10 L145, p14 L248, p16 L321, p22 L605, L606, p23 Table A1, p24 Table A1,

-p3, Table 1: No notes for *, **,etc. can be found at the end of the table.

-p4, Table 1: Don’t italicize subsp.

-p5, Table 1: Don’t italicize “and” and “like”.

-p9, Figure 1: The words are too small to see clearly.

-p10, L154: Change the first “and” to “as well as” will make the sentence more easy to understand.

-p11, Table 2: Italicize “Leishmania” and “Anaplasma boleense”.

-p12, L188 ,190, 193: Use “A.” instead of “An.” for Anaplasma.

-p13, Figure 3: The species names should be italicized.

-p13, Figure 4: The species names should be italicized.

-p14, L256, 260: Use “B.” instead of “Bo.” for Borrelia.

-p14, Figure 5: The species names should be italicized.

-p14, Figure 5: Please omit the additional comma.

-p20, Table 3: Names of species should be italicized.

-p22, L591-593: If the authors don’t want to acknowledge any people, I think they can omit this section.

-p23, 24, Table A1: Don’t italicize spp.

-p25, L629: Italicize “Ehrlichia”.

-p25, Table A2: Don’t italicize “Panola mountain” and spp.

-p25, L662: Don’t italicize “Panola mountain”.

-p26, Table A3: Since percentage was noted in the title column, the % can be omitted, and only showed the number in the parentheses.

-p29, L728: Don’t italicize spp.

-p29, Table A6: The notes should be superscript.

-p29-35, References: Different formats were presented in the section. Please check and modify them.

Author Response

We thanks the reviewer for his valuable comments. We tried to follow all the recommendations and performed the corrections as suggested.

-p3, Table 1: Were all these primers and probes designed by the authors in the study? Or any of them were adopted from previous publications?

  • One part of the primers came from a previous study (Michelet et al., 2014), and the other part were newly design for this study. This information has been added to the Table legend line 107-110 “*: Design from Michelet et al., 2014 [18]. **: include all the controls belonging to the genus described in the table and targeted by specific design. Plasmids used as control are recombinant PBluescript IISK+ containing the target gene.”

-p11, L172: B. ovis? or B. bovis? According to the description (p10, L144-145), Borrelia burgdorferi sensu stricto, Babesia ovis, and Rickettsia rickettsii were removed from the high throughput screening in the study because of the failure in detecting positive control or low specificity.

  • Line 177 : B. ovis has been replaced by B. bovis accordingly.

-p11, L175-176: Rickettsia spp. detected in 15.7-23.5% of the R. micropolus samples...Figure 2 shows 15.2-23%.

-p12, L189: 3.6-4.8%? Figure 2 shows 3-4.2%.

-p12, L201: 42.3%? Figure 2 shows 43.3%. 24.1-31.9%? Figure 2 shows 23.6-31.5%.

-p12, L211: 4.2-6.6%? Figure 2 shows 4.2-6.7%.

  • All the percentages have been corrected accordingly to figure 2.

-p13, L227: Please provide the number of the ticks in the parenthesis, e.g. N/n.

  • Information added Line 232 “In addition, in around 50% (at least 4/8 ticks) and 18% (at least 22/114 ticks) of the R. microplus specimens positive…”
  • Nb: here “at least” considering the lower infection rate in pools of R. microplus.

-p14, L246: 5.1% or 5.3% (in Figure 2)?

  •  Percentage have been corrected accordingly to figure 2.

Other comments:

-p2, L67: However, very few information are available...Information is an uncountable noun.

  •  Corrected accordingly.

-p3, L104: sensu lato. The Latin referent words and phrases are usually italicized. This also happened in p3 Table 1 (Infected Rhipicephalus sanguineus s. l.), p4 (Borrelia burgdorferi sensu stricto), p8 (Rhipicephalus sanguineus s. l.), p8 L113, p10 L145, p14 L248, p16 L321, p22 L605, L606, p23 Table A1, p24 Table A1,

  • The font of “s.l., s.t., sensu lato, sensu stricto“ has been corrected.

-p3, Table 1: No notes for *, **,etc. can be found at the end of the table.

  • The legend has been added to the Table line 107-110 “*: Design from Michelet et al., 2014 [18]. **: include all the controls belonging to the genus described in the table and targeted by specific design. Plasmids used as control are recombinant PBluescript IISK+ containing the target gene.”

-p4, Table 1: Don’t italicize subsp.

  • Corrected as requested.

-p5, Table 1: Don’t italicize “and” and “like”.

  • Corrected as requested.

-p9, Figure 1: The words are too small to see clearly.

  • Figure 1 has been replaced by an image of better quality.

-p10, L154: Change the first “and” to “as well as” will make the sentence more easy to understand.

  • Corrected as requested Line 156

-p11, Table 2: Italicize “Leishmania” and “Anaplasma boleense”.

  • Leishmania” has been italicized. However, as the term "Candidatus" prevent the use of Italic for the following genus/species name, “Candidatus Anplasma boleense” has been kept as it in the paper.

-p12, L188 ,190, 193: Use “A.” instead of “An.” for Anaplasma.

  • Corrected as requested.

-p13, Figure 3: The species names should be italicized.

  • Corrected as requested.

-p13, Figure 4: The species names should be italicized.

  • Corrected as requested.

-p14, L256, 260: Use “B.” instead of “Bo.” for Borrelia.

  • Corrected as requested.

-p14, Figure 5: The species names should be italicized.

  • Corrected as requested.

-p14, Figure 5: Please omit the additional comma.

  • Corrected as requested.

-p20, Table 3: Names of species should be italicized.

  • Corrected as requested.

-p22, L591-593: If the authors don’t want to acknowledge any people, I think they can omit this section.

  • Thank you for seeing this, the section has been updated.

-p23, 24, Table A1: Don’t italicize spp.

  • Corrected as requested.

-p25, L629: Italicize “Ehrlichia”.

  • Corrected as “Panola Mountain Ehrlichia” among the paper as requested.

-p25, Table A2: Don’t italicize “Panola mountain” and spp.

  • Corrected as “Panola Mountain Ehrlichia” among the paper as requested.

-p25, L662: Don’t italicize “Panola mountain”.

  • Corrected as “Panola Mountain Ehrlichia” among the paper as requested.

-p26, Table A3: Since percentage was noted in the title column, the % can be omitted, and only showed the number in the parentheses.

  • “%” removed as requested.

-p29, L728: Don’t italicize spp.

  • Corrected as “Panola Mountain Ehrlichia” among the paper as requested.

-p29, Table A6: The notes should be superscript.

  • Corrected as requested.

-p29-35, References: Different formats were presented in the section. Please check and modify them.

  • References checked and modified accordingly.

Round 2

Reviewer 1 Report

Revisions adequate.  Nice work.